# Forensic Microbiology: Challenges in Detecting Sexually Transmitted Infections

**DOI:** 10.3390/diagnostics15101294

**Published:** 2025-05-21

**Authors:** Ana Eira, Jennifer Fadoni, António Amorim, Laura Cainé

**Affiliations:** 1National Institute of Legal Medicine and Forensic Sciences, I.P., Centre Branch, 3000-548 Coimbra, Portugal; ana.c.eira@inmlcf.mj.pt (A.E.); antonio.j.amorim@inmlcf.mj.pt (A.A.); 2National Institute of Legal Medicine and Forensic Sciences, I.P., North Branch, 4050-202 Porto, Portugal; jennifer.n.fadoni@inmlcf.mj.pt; 3LAQV&REQUIMTE, Laboratory of Applied Chemistry, Department of Chemical Sciences, Faculty of Pharmacy, University of Porto, 4050-313 Porto, Portugal; 4Faculty of Sciences, Lisbon University, 1749-016 Lisboa, Portugal; 5Faculty of Medicine, Lisbon University, 1649-028 Lisboa, Portugal; 6Faculty of Medicine, Porto University, 4200-319 Porto, Portugal

**Keywords:** sexual crimes, sexually transmitted infections, detection methods, gonorrhoea, chlamydia, syphilis, trichomoniasis

## Abstract

Sexual assault crimes consist of acts committed without consent and represent a major global issue with serious implications for victims. These acts have both short- and long-term consequences on the physical, mental, and sexual health of victims, with sexually transmitted infections (STIs) being one of the direct outcomes of such crimes. Sexually transmitted infections constitute a serious global public health problem and can lead to severe consequences. These infections may be caused by bacteria, viruses, or parasites and are transmitted through sexual contact. Some of the most common STIs among the general population and victims of sexual crimes include gonorrhoea, chlamydia, trichomoniasis, and syphilis. In most carriers, these infections are asymptomatic, making their detection particularly challenging. Considering the importance of further research in this field, the primary objectives of this study are to review the existing literature on the incidence of major STIs in victims of sexual crimes, to identify the various risk factors associated with these infections, and to explore their public health implications. Additionally, this study aims to assess different STI detection techniques, analyzing their advantages and disadvantages. Studies on this topic are crucial for better understanding the role of sexually transmitted infections in the context of sexual crimes. However, throughout this work, it was verified that point-of-care methods are a good option to allow the diagnosis to be faster and more accurate, when compared to other methods of detecting sexually transmitted infections.

## 1. Introduction

Sexual violence is a global issue that affects various aspects of health, including medical and psychological domains [1]. It is both a public health and human rights concern and can be perpetrated against both men and women, although women are at a higher risk [2].

In forensic science, the accurate and timely identification of sexually transmitted infections (STIs) plays a crucial role in the reconstruction of criminal events, particularly in cases involving sexual assault. Forensic microbiology has emerged as a key interdisciplinary domain that bridges traditional microbiological diagnostics with legal investigations. The ability to detect pathogens in biological samples collected from crime scenes or victims can provide essential corroborative evidence, aiding in suspect identification, timeline estimation, and the establishment of contact or transmission pathways. Although the exact prevalence of sexual violence remains unknown, it is a reality for many individuals [1]. According to the World Health Organization (WHO), “Over the course of their lives, one in three women—around 736 million people—are subjected to physical or sexual violence by their partner or sexual violence by a non-partner” [3]. This high incidence necessitates effective strategies for both medical intervention and forensic investigation.

Sexual assault has serious short- and long-term impacts on the victim’s physical, sexual, and reproductive health [2,4]. Among the various associated implications, sexually transmitted infections are a direct consequence of sexual crimes. These infections, caused by bacteria, viruses, or parasites, can be transmitted through body fluids or interaction with the skin through vaginal, oral, or anal sex [5]. They affect individuals of all ages, genders, or sexual orientations and can lead to serious health complications if left untreated [6].

The global incidence of STIs varies widely and remains poorly understood for several reasons. Many infections are asymptomatic, diagnostic procedures are scarce in the most affected regions, and surveillance programs are either absent or underdeveloped in many parts of the world [5]. Additionally, inaccuracies in the epidemiological tracking of STIs further hinder the determination of their true prevalence [7].

The prevalence of these infections varies across countries due to economic differences, cultural perspectives, and the lack of education and prevention of STIs [7].

Currently, four sexually transmitted infections have the known highest prevalence and are the most widely studied worldwide: gonorrhoea, caused by the bacterium *Neisseria gonorrhoeae*; chlamydia, caused by the bacterium *Chlamydia trachomatis*; syphilis, caused by the bacterium *Treponema pallidum*; and trichomoniasis, resulting from infection with the protozoan *Trichomonas vaginalis*. These infections have significant lifelong consequences for victims, making the development of effective, low-cost, and rapid detection methods crucial to facilitate timely diagnosis. The methods used for detection can be direct or indirect, and for each type of infection, different techniques are studied to determine the most suitable approach.

The main aims of this article are to review the existing literature on the incidence of major sexually transmitted infections in victims of sexual crimes, to identify the various risk factors associated with these infections, and to explore their implications for public health. Additionally, this work aims to assess different STI detection techniques, analyzing their advantages and disadvantages, to enhance the quality and effectiveness of care provided to victims.

This review focuses on the most prevalent sexually transmitted infections (STIs) and the available diagnostic tools used for their detection, particularly in the context of sexual crimes. While these infections are clinically significant on their own, their identification may also serve as supporting evidence in forensic investigations, helping to confirm sexual contact, assess the time of infection, and support or refute allegations of abuse. The aim is to provide a concise overview of the main STIs, their symptoms, and diagnostic techniques, with emphasis on their forensic applicability.

## 2. Methods

The analysis of detection methods considered laboratory tests such as polymerase chain reaction (PCR), serological tests, and rapid tests, focusing on their sensitivity, specificity, and effectiveness in the context of sexual crime victims. The present study is a narrative review; therefore, no formal criteria for study selection were employed.

In November 2024, a search was conducted on the PubMed, Scopus, and Google Scholar databases. The search included the keywords “sexual crimes, sexually transmitted infections, detection methods, Gonorrhoea, Chlamydia, Syphilis, Trichomoniasis”, and combinations of these terms were used to refine the search.

The selected articles included clinical trials, observational studies (cohort and cross-sectional), and systematic reviews, with no temporal or linguistic restrictions, that addressed the relationship between sexual crimes and STIs, as well as the methods used for their detection, with a particular emphasis on those applied in forensic settings.

Although a single gold standard test was not formally defined for all infections addressed in this review, it is important to note that molecular diagnostic methods—particularly nucleic acid amplification tests (NAATs), including polymerase chain reaction (PCR) and transcription-mediated amplification (TMA)—alongside culture techniques (e.g., cell culture for *Chlamydia trachomatis* or selective media for *Neisseria gonorrhoeae*), are widely recognized in the scientific literature as reference or gold standard methods. The sensitivity and specificity values presented in this review are those reported in original studies that used such reference methods as comparators to assess the performance of alternative or point-of-care (POC) tests. These comparisons are standard practice in diagnostic accuracy studies, especially in the absence of a universally applicable gold standard across all clinical and forensic settings.

Although this is a narrative review and no formal systematic exclusion criteria were applied, studies were excluded if they did not address the association between sexually transmitted infections and sexual crimes, or if they lacked methodological rigor or relevance to the analysis of diagnostic methods. Preference was given to studies that provided data on diagnostic performance and that were applicable to forensic or clinical contexts.

## 3. Sexual Violence and STI Implications

The WHO has defined sexual violence as “any sexual act, attempt to consummate a sexual act, or other act directed against a person’s sexuality through coercion (…) regardless of their relationship with the victim and in any context. (…) defined as the penetration by physical or other coercion of the vulva or anus with a penis, other part of the body or object” [3]. It often involves physical force or psychological manipulation, particularly in contexts of power imbalance, such as among individuals with disabilities, adolescents, incarcerated persons, and those in institutional or conflict settings [1,8].

According to the World Health Organization, nearly 30% of women globally have experienced physical and/or sexual violence by an intimate partner or non-partner during their lifetime. Sexual violence can result in a range of physical, reproductive, psychological, and social consequences, including injuries, unwanted pregnancies, sexually transmitted infections, and long-term mental health disorders. Table A1 summarizes the potential consequences across these domains [3,9].

To better understand the global impact of STIs in the context of sexual violence, Table 1 summarizes WHO-estimated prevalence rates across different regions.

As sexual violence is a major risk factor for the acquisition of STIs, understanding its global distribution is essential. Figure 1 shows the estimated lifetime prevalence of intimate partner violence by region, based on WHO data. However, it is important to note that this figure does not reflect the full spectrum of sexual violence, particularly assaults committed by non-intimate partners. Data on non-intimate partner sexual violence remain scarce and less systematically recorded across global health systems, despite their relevance in forensic and clinical contexts. This gap underscores the need for improved surveillance and reporting mechanisms that capture all forms of sexual violence, regardless of the relationship between victim and perpetrator.

## 4. Sexually Transmitted Infections (STIs)

Often, when STIs are detected in adults, they may be used as evidence of sexual assault or to question the victim’s previous sexual behavior. In the case of children, the presence of an STI can be a crucial indicator of sexual violence or may help establish a link between the perpetrator and the victim [10].

Globally, there are approximately 1 million new STI cases every day, with an estimated 374 million new infections annually, including chlamydia, gonorrhoea, syphilis, and trichomoniasis. The prevalence of STIs varies depending on the specific infection and the economic conditions of each country, typically being higher in regions with less-resourced healthcare systems [10].

The WHO [3] reports an increasing number of untreated STI cases worldwide due to treatment failures and the fact that most infections remain asymptomatic in the majority of affected individuals. Although most STIs are non-fatal, they can lead to severe health complications. The greatest risks associated with STIs stem from unprotected sex, an increase in the number of sexual partners, and the emergence of new sexual partners [11].

While global prevalence data for sexually transmitted infections are well established, specific data linking STI incidence directly to cases of sexual assault are limited and often underreported. This is partly due to the sensitive nature of sexual violence, variations in reporting practices, and the asymptomatic nature of many STIs. Studies have shown that sexual assault victims are at elevated risk of acquiring STIs due to forced, unprotected intercourse and potential exposure to multiple pathogens. However, quantifying this risk remains challenging due to inconsistent data collection across regions and the overlap with pre-existing infections. The presence of an STI may support allegations of non-consensual sexual contact, particularly when it is newly diagnosed and the timing of transmission can be reasonably inferred. Nonetheless, additional epidemiological research is needed to more precisely estimate the prevalence and patterns of STI transmission specifically attributable to sexual violence.

The following sections (Section 4.1, Section 4.2, Section 4.3 and Section 4.4) present the four most prevalent sexually transmitted infections—gonorrhoea, chlamydia, syphilis, and trichomoniasis—commonly observed in the general population and among victims of sexual assault. For each infection, a brief clinical overview is provided to describe key symptoms and possible complications, followed by a summary of the most relevant diagnostic methods, with emphasis on their applicability in forensic contexts.

### 4.1. Gonorrhoea

Gonorrhoea is a sexually transmitted infection caused by *Neisseria gonorrhoeae*, a Gram-negative diplococcus belonging to the class *Betaproteobacteria* and the family Neisseriaceae [12,13].

This infection is considered a serious global health concern due to the high number of asymptomatic cases, the bacterium’s ability to undergo significant antigenic variation on its surface, and the emergence of antibiotic-resistant strains [7].

*Neisseria gonorrhoeae* is an obligate human pathogen [12] that affects the lower genital tract, pharynx, and rectum, with different complications depending on the sex of the individual. In women, the most commonly affected site is the cervix, whereas in men, it is the anterior urethra. The most prevalent symptoms include purulent cervical or urethral discharge, discomfort, dysuria, urethritis, or cervicitis [7], typically appearing 2 to 10 days after infection [6].

Individuals with rectal infections may experience anal discharge, discomfort, and pain in the affected area. Gonorrhoea is also one of the leading causes of pelvic inflammatory disease (PID), which can result in chronic abdominal pain, dyspareunia, infertility, or ectopic pregnancy [6]. Additionally, this bacterium can be vertically transmitted, meaning it can be passed from the mother to the child during pregnancy, childbirth, or breastfeeding. It has the potential to cause chorioamnionitis, septic abortion, preterm birth, and neonatal conjunctivitis, posing a significant risk to vision [7].

In forensic cases, the identification of *N. gonorrhoeae* in a victim may support allegations of recent sexual contact, particularly when clinical symptoms are present and corroborated by microbiological evidence.

#### 4.1.1. Methods for Detecting *Neisseria gonorrhoeae*

##### Microscopy

Microscopic analysis is a cost-effective method that does not require highly sophisticated equipment and can be performed during the initial medical examination of the victim [14]. This technique is characterized by its high sensitivity and specificity and serves as a rapid diagnostic test [15].

In symptomatic men, microscopy combined with Gram staining enables the identification of diplococci within polymorphonuclear leukocytes, a key criterion for the presumptive diagnosis of gonorrhoea [15]. Gram staining is a rapid differential technique that distinguishes bacteria based on cell wall structure. In this method, *Neisseria gonorrhoeae*, a Gram-negative diplococcus, appears as pink, kidney-shaped cells often located within polymorphonuclear leukocytes. This allows for a quick presumptive diagnosis, particularly in symptomatic male patients.

Higher sensitivity and specificity have been observed in urethral swab samples from symptomatic men, whereas lower sensitivity has been reported in urethral samples from asymptomatic men and endocervical or urethral samples from women. This discrepancy may be attributed to the lower bacterial load in urethral samples and the presence of other bacterial species in endocervical samples [12].

However, this technique has certain limitations, as its accuracy depends on the technician’s expertise, and microscopic examination must be conducted within 10 min of sample collection to ensure reliable results [14].

As an alternative to Gram staining, methylene blue staining has been proposed, demonstrating high sensitivity and specificity in the diagnosis of gonococcal urethritis in men [12].

##### Culture

The technique of culturing microorganisms on specific media demonstrates good sensitivity and specificity. Although it does not yield results for all sample types, it is effective for urethral and cervical samples, whereas conjunctival, rectal, and oropharyngeal samples require optimal growth conditions for successful culture [15].

For the detection of this bacterium, the primary species must be cultured in the medium of non-selective chocolate agar and selective agar that have antimicrobial agents that inhibit the growth of bacteria and fungi. Culture media that have incorporated antibacterial and antifungal agents are Thayer–Martin, Martin Lewis, and New York City [15].

This technique has evolved over time, enhancing its efficiency and precision, particularly with the integration of MALDI-TOF mass spectrometry [14]. This method allows for the direct identification of bacteria from agar plates, generating a unique spectral profile that correlates with the specific microorganism [15]. Another method is Vitek 2 (BioMérieux, Marcy-l’Étoile, France), an automated microbiological identification system based on biochemical and enzymatic profiling, which allows for the rapid identification of a variety of bacterial and fungal species, including *Neisseria* species [16].

However, one of the main drawbacks of this technique is its relatively slow process, which can also be costly in certain cases. Additionally, *Neisseria gonorrhoeae* is a fragile bacterium, and its culture may exhibit low sensitivity due to inadequate transportation conditions [14]. Certain transport media have been developed to maintain bacterial viability for up to 48 h at room temperature [17].

##### Nucleic Acid Amplification Tests (NAATs)

Compared to culture, NAAT (Nucleic Acid Amplification Test)—a group of molecular diagnostic methods that amplify the genetic material of pathogens—offers higher sensitivity and can be applied to a larger number of samples. NAATs include techniques such as polymerase chain reaction (PCR), loop-mediated isothermal amplification (LAMP), transcription-mediated amplification (TMA), and strand displacement amplification (SDA) while requiring less stringent conditions for transport and storage [17], as they do not necessitate viable organisms for detection [14].

One of the key advantages of this technique is its rapid response time and higher sensitivity compared to other methods. Additionally, this procedure can be automated, performed by individuals with minimal experience, and multiplexed, allowing for the simultaneous detection of multiple microorganisms from a single sample [14].

PCR-based assays

Different types of PCRs, including single-step, nested, real-time, singleplex, and multiplex PCR, are utilized to simplify diagnosis and reduce turnaround time. This technique exhibits very high sensitivity and specificity and is particularly valuable, as it enables the detection of multiple microorganisms within a single sample.

However, its main limitations stem from the requirement for specialized instruments and laboratory facilities, which may not be readily available in all settings [14].

Isothermal and microfluidic amplification tests

Compared to PCR-based tests, this technique is faster and more cost-effective for detecting STIs. In this process, DNA/RNA (deoxyribonucleic acid/ribonucleic acid) amplification occurs at a lower and constant temperature without requiring thermal cycling, thereby reducing the time needed to obtain results. Additionally, results can be observed and evaluated visually by detecting turbidity or color changes in the reaction tube [14].

This technique encompasses several molecular diagnostic methods that involve various chemical reactions. The primary methods used for STI detection include loop-mediated isothermal amplification (LAMP), transcription-mediated amplification (TMA), strand displacement amplification (SDA), helicase-dependent amplification (HDA), and recombinase polymerase amplification (RPA). Furthermore, the enzymes utilized in these methods are less affected by inhibitory substances present in the samples, which reduces both the time and cost associated with sample preparation and extraction [14].

However, these techniques also present certain limitations. False positives may arise during amplification, primer design for detecting bacterial genetic sequences can be challenging, and the final amplification products often vary in length, making them difficult to use in subsequent applications [18].

##### Sensitive Detection Without Amplification

To minimize sample handling and processing time, alternative molecular methods have been developed that allow direct detection of DNA or RNA from clinical samples without requiring amplification. One example is microwave-accelerated metal-enhanced fluorescence (MAMEF), which extracts nucleic acids via low-power microwaves and uses enhanced fluorescence for detection [14].

It is important to note that, although these methods involve nucleic acid detection, they do not fall under the category of NAATs, as they do not include an amplification step. As such, they represent a distinct class of diagnostic tools that may be suitable for point-of-care use, though further development is needed to adapt these techniques to complex biological matrices such as serum or whole blood.

However, a major limitation of this method is that it is currently validated only for assays involving synthetic or laboratory-prepared samples suspended in buffer solutions. Its application to real biological specimens—such as vaginal swabs, serum, whole blood, or saliva—remains limited and requires further development [19].

This method demonstrates good sensitivity and concordance with PCR, making it a promising approach for use in point-of-care (POC) diagnostic tests [14].

##### Immunological Tests

This technique allows the detection of antigens or the presence of antibodies produced by the immune system at the time of infection, with lateral flow immunochromatographic tests being used [14]. When infection caused by the bacterium *Neisseria gonorrhoeae* occurs, there is an increase in IgG and IgA, and these antibodies are detected using immunological methods [20].

These tests exhibit high specificity and can provide results within 25 to 40 min; however, their sensitivity is highly variable, often being low in many cases.

This issue is common across multiple sexually transmitted infections (STIs), and the WHO does not recommend their use until more effective diagnostic methods become available.

The main disadvantages of this method include the visual interpretation of results, which may lead to operator errors, as well as the potential for detecting past infections, rather than active ones [14].

### 4.2. Chlamydia

*Chlamydia trachomatis* (CT) is a gram-negative, obligate intracellular bacterium [21,22] that belongs to the order *Chlamydiales* within the *Chlamydiaceae* family [23]. Humans are its only natural host [24], and it is the causative agent of chlamydia.

Chlamydia is often an asymptomatic infection, which contributes to a higher risk of transmission [19]. In symptomatic cases, the incubation period typically ranges from 1 to 3 weeks [25].

In women, this infection can lead to long-term complications, affecting both the lower and upper reproductive systems, including the uterus, fallopian tubes, and ovaries. Potential consequences include chronic pelvic pain, pelvic inflammatory disease, an increased risk of ectopic pregnancy, and neonatal complications through vertical transmission, such as conjunctivitis and/or pneumonia. Additionally, it may result in infertility, gynecological tumors [21], endometritis, or peri-hepatitis [24].

In men, *C. trachomatis* infection can cause urethritis [21] and epididymitis, which may lead to fever, scrotal pain, and swelling. Prostatitis can result in pain during and after sexual intercourse, chills, painful urination, and lower back pain [26]. Furthermore, this infection can contribute to male infertility [21], orchiepididymitis, lymphogranuloma venereum [25], ocular infections such as conjunctivitis, and can also affect the throat and rectum [26].

From a forensic perspective, chlamydial infection can be an important marker of sexual exposure, especially in children or individuals unable to provide informed consent, where its detection may strongly suggest abuse.

#### 4.2.1. Methods for Detecting *Chamydia trachomatis*

##### Culture

This technique requires viable bacteria, and its detection rate ranges from 60% to 80%, even when performed in specialized laboratories by trained professionals [27].

For the detection of this bacterium, the culture method is carried out in cell lines capable of supporting its intracellular growth. The most commonly used are McCoy, HeLa cells, a human epithelial cell line derived from cervical cancer, and BGMK (Buffalo Green Monkey Kidney) cells, a non-human primate epithelial cell line. These cell types provide a suitable environment for the propagation of *Chlamydia trachomatis* [28].

Due to the rigorous care required in the collection, transportation, and processing of samples, this method presents significant disadvantages, as these factors can interfere with its sensitivity, which may vary between 70% and 80%. Other drawbacks include its labor-intensive nature [29] and time-consuming process, as results may take 3 to 7 days to be obtained [28].

Additionally, toxic substances may sometimes be present in clinical samples, and commensal microbes can overgrow, further complicating the analysis [27].

##### Antigen Detection Methods

In this technique, bacterial viability is not required, and no specific precautions are necessary for sample collection and transport, unlike culture-based methods [29].

A direct fluorescent antibody (DFA) has been developed for direct application to clinical samples. This technique utilizes fluorescein isothiocyanate-labeled monoclonal antibodies that bind to Chlamydia lipopolysaccharide (LPS) or the major outer membrane protein (MOMP). The use of MOMP-specific antibodies has significantly improved specificity to 98–99% and sensitivity to 80–90% compared to culture methods. Additionally, this method is rapid, with a turnaround time of approximately 30 min [29].

Another developed test is the enzyme immunoassay (EIA), which detects specific LPS antigens that are more abundant and soluble than MOMP [29]. Detection can be performed using either monoclonal or polyclonal antibodies; however, this technique requires a specialized technician [30]. The manual procedure typically takes around 3–4 h [28]. A major drawback of this technique is that LPS-specific antibodies can cross-react with LPS from other bacteria, leading to false-positive results [29].

Several EIA-based commercial tests are available, including Chlamydiazyme (Abbott Diagnostics, North Chicago, IL, USA) and Microtrak EIA (Behring, Palo Alto, CA, EUA) [28].

##### Serological Test

The serological methods developed for detecting Chlamydia infections include the microimmunofluorescence (MIF) test and the enzyme-linked immunosorbent assay (ELISA), which allow the detection of the IgG antibody that is produced during an acute infection caused by the bacteria [28].

The MIF test has been identified as the most sensitive method, capable of providing specific data. However, it has not been widely adopted as a diagnostic tool due to its limitations and high cost [29]. Additionally, this method is highly time-consuming and labor-intensive, and fluorescence signal interpretation is prone to errors [27].

Conversely, the ELISA test, which utilizes lipopolysaccharide (LPS) from the *Chlamydia* genus, can yield false-positive results due to the possibility of cross-reactions with other *Chlamydia* species. Furthermore, as most antibodies produced during infection remain in circulation, a positive result does not necessarily indicate an active infection [29].

##### Microchip-Based Multiplexed Immunoassay

This type of method enables the simultaneous performance of multiple immunoassays from a small vaginal sample. In this context, the liposomal nanovesicle technique could facilitate the development of a signal amplification system, thereby increasing the sensitivity of this type of assay [31].

This method utilizes small amounts of purified, high-density proteins, which are immobilized on a microscope slide. Specific antibodies are then introduced, and their presence is detected through fluorescent markers or radioisotopes, provided that the antibody binds to the target protein [32].

Multiplex immunoassays are characterized by high clinical sensitivity, as they are capable of detecting antibodies or antigens in patient samples with high accuracy, even when only small sample volumes are available. When an inflammatory disease is triggered by multiple pathogens, it is crucial to detect all of them within a single reaction, a capability provided by multiplex methods [31].

Multiplex PCR techniques enable the simultaneous detection of multiple pathogens from a single clinical sample. In the context of STIs, this approach is particularly valuable for screening infections such as *Neisseria gonorrhoeae*, *Chlamydia trachomatis*, and *Trichomonas vaginalis* in one reaction, improving diagnostic efficiency and reducing the need for multiple tests (Figure 2 and Figure 3).

##### Molecular Techniques

One of the methods developed for the detection of Chlamydia infections is the nucleic acid hybridization (NAH) technique, with the PACE 2 test (Gen-Probe, San Diego, CA, USA) being the most widely used [29]. The PACE 2 test is a nucleic acid hybridization assay that uses a chemiluminescent DNA probe to detect ribosomal RNA specific to *Chlamydia trachomatis*. The PACE 2C version also allows simultaneous detection of *Neisseria gonorrhoeae*. These tests provide rapid and specific results without the need for nucleic acid amplification [27,29]. The procedure for this test takes approximately 2–3 h, and the technical expertise required is comparable to that needed for enzyme immunoassay (EIA) tests, although automation is possible [28]. These tests provide rapid and specific results without the need for nucleic acid amplification.

Another nucleic acid amplification test (NAAT) test, Hybrid Capture II (Digene Corporation, Gaithersburg, MD, USA), incorporates a signal amplification component to enhance sensitivity [29].

Further advancements in nucleic acid amplification testing (NAAT) have led to the development of polymerase chain reaction (PCR) and ligase chain reaction (LCR) methods for the diagnosis of Chlamydia infections [29]. These nucleic acid amplification techniques, such as PCR and TMA, exhibit high analytical sensitivity, enabling the detection of minute quantities of bacterial DNA or RNA in clinical samples, often down to a few copies per reaction. In real-world settings, they have also demonstrated high clinical sensitivity, effectively identifying infected individuals, including those with low bacterial loads or asymptomatic cases. When combined with automated nucleic acid extraction, results can be obtained within a few hours [27].

The sensitivity of these methods has been further enhanced through the implementation of pre-analytical steps involving coated magnetic beads, which improve both the quantity and quality of nucleic acid separation [27].

Following these advancements, new molecular technologies have emerged for Chlamydia detection, including TMA-based (transcription-mediated amplification) tests such as APTIMA Combo 2 (Gen-Probe Inc., San Diego, CA, USA) and ProbeTec (BD Diagnostic Systems, Franklin Lakes, NJ, USA) [29]. NAATs (Nucleic Acid Amplification Tests) specifically designed to detect *C. trachomatis* plasmid DNA include the COBAS Amplicor *C. trachomatis* test (Roche Diagnostic Systems, Basel, Switzerland), which employs PCR that has synthetic oligonucleotide primers that bind to specific sequences of this bacterium, and the BDProbeTec™ ET test (Becton Dickinson, NJ, USA), which utilizes SDA (strand displacement amplification) [30].

Another NAAT method, the Cepheid Xpert CT/NG (Sunnyvale, CA, USA), is a rapid and user-friendly real-time PCR assay. It operates within a closed system, requires minimal operator intervention, and delivers quick and efficient results [29].

Despite their advantages, these techniques also present certain limitations. They require trained professionals and specialized laboratory equipment, making them costly and difficult to implement in resource-limited settings [29]. Additionally, these methods may cross-react with other *Chlamydia* species and can be affected by transport conditions [31].

### 4.3. Syphilis

#### 4.3.1. Overview and Clinical Manifestations

Syphilis is a multisystemic sexually transmitted infection [33] caused by the bacterium *Treponema pallidum*, a spirochete that affects only humans and cannot be cultivated in vitro [34]. It is transmitted primarily through direct contact with infectious mucocutaneous lesions or via vertical transmission from mother to fetus during pregnancy [35]. The disease progresses through four clinical stages—primary, secondary, latent, and tertiary—each with distinct characteristics [36].

**Primary syphilis** typically presents as a painless ulcer or chancre at the site of inoculation, appearing around 21 days after exposure. It may go unnoticed and spontaneously heals within 3–6 weeks. Regional lymphadenopathy is often presented [34,35,36].**Secondary syphilis** occurs 3 to 12 weeks later and is characterized by a widespread maculopapular rash, mucocutaneous lesions, lymphadenopathy, and systemic symptoms such as fever, malaise, and arthralgia. This stage remains highly infectious and may resolve without treatment [34,35].**Latent syphilis** is an asymptomatic stage that follows the resolution of primary and secondary symptoms. It is divided into early latent (<1 year post-infection) and late latent (>1 year). The disease remains serologically detectable but is not transmissible sexually [35].**Tertiary syphilis** may develop 10–20 years after the initial infection in untreated individuals. It is associated with severe complications including cardiovascular lesions (e.g., aortitis), gummatous skin or bone lesions, and neurosyphilis [35,36,37,38].**Neurosyphilis** can occur at any stage and may be asymptomatic or present with neurological and ophthalmic manifestations, such as cranial nerve dysfunction, meningitis, stroke, and progressive dementia. Parenchymatous neurosyphilis may also lead to psychiatric symptoms, motor disturbances, and sphincter dysfunction [34,35,39].**Congenital syphilis** results from transplacental transmission and can lead to miscarriage, stillbirth, or severe neonatal complications. It is classified as early (manifesting before age 2) or late (after age 2), with clinical signs ranging from hepatosplenomegaly and skin lesions to sensorineural deafness and dental abnormalities [40].

The presence of syphilitic lesions or serological evidence in alleged victims of sexual assault can support forensic conclusions, particularly when infection is newly acquired and correlated with other findings.

Syphilis progresses through distinct clinical stages—primary, secondary, latent, and tertiary—each with characteristic symptoms and varying degrees of severity. A summary of the signs and complications associated with each stage is provided in Table A2. These stages may include painless chancres, mucocutaneous lesions, neurological and cardiovascular complications, and, in untreated cases, irreversible damage. Congenital and neurosyphilis can also occur and pose significant diagnostic and clinical challenges.

#### 4.3.2. Methods for Detecting *Treponema pallidum*

##### Animal Inoculation Test

This type of test utilizes various animal models to study the infection capacity of *T. pallidum*, with rabbits being the most commonly used, as the resulting pathological changes can be observed macroscopically. The test demonstrates a sensitivity and specificity of 100% [41].

Samples for this test can be used as long as they have been collected within one hour or have been immediately frozen and stored in liquid nitrogen at temperatures of −78 °C or lower [42]. However, this method has several disadvantages, including the requirement for specialized technicians, ethical concerns due to the use of animals, and a lengthy processing time [41].

##### Histopathological Staining

In this method, tissue samples are fixed with formalin and stained using silver stain or immunohistochemistry, enabling the visualization of spirochetes or treponemes [34]. However, when silver staining techniques, including Warthin–Starry staining, are applied to tissue from primary or secondary syphilis lesions, the results can be challenging to interpret due to low sensitivity and specificity.

In contrast, immunohistochemical staining, which utilizes fluorescent antibodies against *T. pallidum*, is a more sensitive and highly specific method for detecting the bacterium [34,43].

##### Microscopy

Dark-field microscopy is particularly useful for early detection of *T. pallidum*, allowing direct observation of its morphology. Detection must be performed within 30 min of sample collection, as bacterial motility is the key characteristic for diagnosis [41].

A positive result provides specific and direct confirmation of syphilis. However, a negative result does not exclude infection, as the bacterial load may be too low for detection, the lesion may be in the healing stage, or an excessive number of surrounding cells may interfere with visualization. The sensitivity and specificity of this technique range from 75% to 100% and 94% to 100% for primary lesions, respectively, while for secondary lesions, sensitivity ranges from 58% to 71%, with a specificity of 100% [44].

Despite its diagnostic value, this method has limitations, including the need for a specialized microscope and trained personnel [34]. Furthermore, it is not recommended for routine diagnosis, as the samples are highly sensitive to light variations and the presence of extraneous particles, which may affect accuracy [41].

##### Direct Fluorescence Antibody (DFA)

In this method, the antigen–antibody interaction is detected using fluorescence with fluorescein isothiocyanate (FITC), producing a mahogany–green coloration, without requiring live bacteria [41]. This technique can differentiate pathogenic from non-pathogenic treponemes by detecting specific antigen–antibody reactions [42].

Monoclonal antibodies used in this method offer a sensitivity range of 73–100% and a specificity of 100%, while polyclonal antibodies provide a sensitivity of 86–90% and a specificity of 96–97% [41].

However, this technique has several drawbacks, including high costs and the fragility of fluorescent materials, which are easily degraded [41]. Additionally, it requires specialized equipment and trained personnel [34] and has a longer turnaround time, with results typically obtained within 1–2 days [44].

##### Polymerase Chain Reaction (PCR)

The PCR-based tests developed for syphilis detection include conventional PCR, nested PCR, real-time PCR, and multiplex PCR, all of which offer rapid results [45].

A study conducted by Matt Shields [46] demonstrated that conventional PCR has a sensitivity ranging from 84.6% to 89.1% and a specificity of 93.1% to 100% for primary syphilis cases. However, for secondary syphilis, the sensitivity decreases to 50%, indicating that conventional PCR is most suitable for detecting primary syphilis. In contrast, nested PCR offers higher specificity and sensitivity compared to conventional PCR, as it utilizes a probe that enhances amplification accuracy, achieving a specificity of up to 95%, though its sensitivity remains below 70% [45].

Real-time PCR (qPCR) enables both detection and quantification of pathogen DNA by measuring fluorescence in real time during amplification. Using a standard curve derived from known DNA concentrations, the initial quantity of target DNA in a sample can be estimated, providing an indirect measure of infection load. However, the accuracy of quantification depends on the sample type and DNA extraction process. Despite this limitation, real-time PCR is widely used due to its speed and ease of implementation in laboratory settings [45].

Multiplex real-time PCR enables the simultaneous detection of multiple pathogens and their quantification without cross-interference, making it particularly useful for individuals with suspected co-infections. This technique reduces costs and detection time by allowing multiple amplifications to be monitored in a single reaction [45]. Additionally, a reverse transcriptase PCR test has also been described for syphilis detection [44].

##### Serological Test

Serological tests for syphilis are divided into two categories: treponemal and non-treponemal tests. Treponemal tests detect antibodies that specifically target *Treponema pallidum* antigens and are generally used to confirm infection, while non-treponemal tests detect antibodies produced in response to cellular damage caused by the infection. The latter are often used for screening and monitoring disease activity or treatment response.

Non-treponemal tests

Non-treponemal tests include the Venereal Disease Research Laboratory (VDRL) test, toluidine red unheated serum test (TRUST), unheated serum reagin (USR), and rapid plasma reagin (RPR) test, with antibodies typically detectable six weeks after infection [41].

These methods rely on detecting tissue damage caused by syphilis through the production of antigens for cardiolipin, cholesterol, and lecithin—common components of human cells [34]. This enables the detection of immunoglobulin M (IgM) and immunoglobulin G (IgG), as well as antibodies produced in response to *T. pallidum* infection [47]. However, false-negative results may occur, particularly in early- or late-stage infections or due to the prozone phenomenon. The latter arises when high antibody concentrations interfere with antigen–antibody binding, which is essential for reactive results [34].

Non-treponemal tests offer several advantages, including widespread availability, low cost, applicability to various sample types, and utility in monitoring treatment effectiveness. However, they also have limitations, such as low sensitivity and the potential for prozone or false-positive results [42].

RPR tests use charcoal particles to facilitate the detection of flocculation, allowing macroscopic analysis of the antigen–antibody complex, while TRUST tests utilize toluidine red dye for visualization. Although most non-treponemal tests are manually performed, some have been automated to improve efficiency and reliability [44].

Treponemal tests

This type of test quantifies the levels of IgM and IgG antibodies that bind to *T. pallidum* proteins. It includes fluorescent treponemal antibody absorption assays (FTA-Abs), agglutination tests such as *T. pallidum* particle agglutination assay (TPPA) and *T. pallidum* hemagglutination assay (TPHA), enzyme immunoassays (EIAs) for various treponemal antigens, rapid diagnostic tests [41], and the microhemagglutination assay for *T. pallidum* antibodies (MHA-TP) [36].

If primary syphilis is suspected and non-treponemal tests yield non-reactive results, treponemal tests should be conducted, as they have a sensitivity close to 100% in primary syphilis and exceed 95% in secondary and tertiary syphilis, respectively [34]. These values correspond to clinical sensitivity, reflecting the performance of these tests in detecting true-positive cases among symptomatic and asymptomatic patients across different disease stages.

These tests use *T. pallidum* as the antigen and can detect antibodies within 2 to 4 weeks after exposure [33,48]. They are employed to verify the reactivity of non-treponemal tests and to confirm syphilis in cases where non-treponemal tests are non-reactive despite clinical symptoms [42].

Individuals with reactive treponemal test results are likely to remain positive for life, as these tests do not differentiate between past and current infections [34]. Additionally, treponemal tests are more complex and expensive to perform and are not suitable for monitoring treatment effectiveness. However, when both treponemal and non-treponemal tests yield reactive results, the specificity of the diagnosis is significantly increased [42].

### 4.4. Trichomoniasis

This infection is caused by *Trichomonas vaginalis* [49], a protozoan parasite belonging to the *Trichomonadidae* family. The incubation period ranges from approximately 5 to 28 days. *T. vaginalis* is a primitive eukaryote with a carbohydrate and energy metabolism similar to that of anaerobic bacteria. It is an obligate human parasite, lacking the ability to synthesize essential macromolecules such as purines, pyrimidines, and lipids, which it acquires from vaginal secretions or through the phagocytosis of host and bacterial cells [50].

The majority of infected individuals remain asymptomatic, with 84% of women and 77% of men exhibiting no symptoms [51]. This sexually transmitted infection (STI) can be classified as acute, chronic, or asymptomatic, depending on the severity of the infection [52].

In cases of acute infection, symptoms include a frothy, yellow or green mucopurulent discharge. Additionally, small punctate hemorrhagic lesions, known as the “strawberry appearance”, may develop on the vaginal and cervical mucosa, occurring in approximately 2% of cases. Chronic infection, on the other hand, presents with milder symptoms, such as itching, dyspareunia, and scant vaginal discharge mixed with mucus. This stage is particularly significant, as individuals with chronic infection are more likely to transmit the parasite [52].

In men, the infection typically resolves within 10 days. Symptomatic cases may present with mild manifestations, including scant, clear to mucopurulent discharge, dysuria, and a mild itching or burning sensation following sexual intercourse [50]. Moreover, *T. vaginalis* infection has been associated with male infertility, as it can contribute to urethritis, prostatitis, epididymitis, and reproductive complications due to inflammatory lesions or impaired sperm production [53].

Approximately 25–50% of women remain asymptomatic during the first six months of infection. If left untreated, *T. vaginalis* can persist indefinitely in the lower urogenital tract [53].

In symptomatic women, vaginitis may develop, and less frequently, adnexitis, which can trigger inflammatory responses in the genital mucosa. This increases the risk of PID through micro-bleeding and heightens susceptibility to cervical neoplasia [53]. Additional symptoms may include vaginal discharge—often diffuse, malodorous, and yellow-green-dysuria, itching, vulvar irritation, and abdominal pain [51]. Furthermore, it may cause vulvar itching and vaginal odor [54], as well as edema, erythema, and colpitis macularis (commonly referred to as “strawberry cervix”), which results from microscopic cervical bleeding [50,55]. The infection can also impact female fertility and contribute to chronic infection of the reproductive tract [53].

During pregnancy, *T. vaginalis* infection can lead to premature rupture of membranes, preterm birth, and low birth weight. Vertical transmission may occur when the newborn comes into direct contact with the maternal genital tract during childbirth or through exposure to infected maternal fluids. In some cases, the infection may also manifest in the infant’s respiratory tract [53]. Additionally, *T. vaginalis* has been associated with an increased risk of cervical cancer [55].

Although often asymptomatic, the detection of *T. vaginalis* in post-assault examinations may serve as supportive evidence of sexual contact, particularly when identified in conjunction with other sexually transmitted pathogens.

#### 4.4.1. Methods for Detecting *Trichomonas vaginalis*

##### Microscopy

This method relies on the presence of a sufficient number of viable, motile microorganisms in the samples prepared for wet mounting [55]. The motility of *T. vaginalis* is temperature-dependent, as these microorganisms can remain viable for approximately six hours at room temperature in phosphate-buffered saline, with their motility gradually decreasing over time. The samples that can be used for this type of technique are vaginal, cervical and secretion samples from the urethra or prostate [56].

The test should be performed within 10 to 20 min of sample collection, as the microorganisms may lose viability. It is an inexpensive and rapid diagnostic method, with a sensitivity ranging from 60% to 70% [57]. However, sensitivity can vary between 38% and 82%, depending on the inoculum size, and when the concentration of *T. vaginalis* is lower than 10⁴ organisms/mL, detection is not possible [56].

One major limitation of this method is the requirement for viable samples [56], as well as the need for a microscope and a trained technician, making it less accessible in resource-limited settings [58].

##### Culture

To perform this technique, the inoculum size must be at least 10² organisms/mL, and its growth should be easily identifiable [56]. This method enables the detection of approximately 95% of infections, with high sensitivity (85–95%) and specificity (95%) [59].

The most commonly used culture medium is Diamond’s TYI in glass tubes. For the identification of *T. vaginalis*, an incubation period of approximately 2 to 7 days is required [56]. However, this method has several disadvantages, including bacterial contamination, high costs [56], and the extended time required for results. In some cases, culture media are not readily available, and the organisms are sensitive to transport conditions, which may lead to their death [59]. Additionally, since incubation is required for several days before detection, these culture methods are not accessible in all locations [60].

To overcome these challenges, a new method called InPouch (BioMed Diagnostics, White City, OR, USA) was developed. In this technique, samples are placed in a specialized pouch consisting of two chambers [56], the upper chamber is used for sample inoculation under sterile conditions and provides an anaerobic environment suitable for the growth of *T. vaginalis*. After incubation, the lower chamber allows for direct microscopic examination of the culture without the need to transfer the sample, minimizing contamination risk and preserving microorganism motility. This dual-chamber design enhances diagnostic efficiency while maintaining ease of use and safety, allowing the entire culture sample to be analyzed for the presence of *T. vaginalis* [54]. These samples can remain at room temperature for about 30 min before inoculation, and test results are typically obtained within 2 to 5 days [57].

The advantages of this technique include its low sample requirement (only 300 to 500 *T. vaginalis* per milliliter) for successful growth [59]. Additionally, it is a simple method that can be used with urethral samples, allows for direct microscopic analysis through the pouch, and does not require expensive storage conditions [50].

The cell culture method offers even greater sensitivity, enabling the detection of *T. vaginalis* in samples containing as few as 3 organisms/mL. This method requires pre-treatment of samples with antibiotics, with Diamond’s TYI medium serving as a transfer medium before the samples are introduced into cell cultures [56].

##### Staining Methods

The Papanicolaou staining method enables the visualization of *Trichomonas* in cervical smears [60]. This technique has a sensitivity of 60% and a specificity of 95% [57].

Staining methods and molecular probes have also been applied directly to clinical samples. These approaches are rapid and exhibit sensitivity comparable to wet mount preparation. However, they present certain disadvantages, including the requirement for specialized equipment, trained personnel, and high costs [55].

Due to the fixative applied during the staining process, these microorganisms may lose their motility, and *T. vaginalis* may not always display its characteristic pear-shaped morphology. Therefore, this technique is most effective when used in conjunction with direct observation methods, such as wet mount preparation, which allows for the assessment of the organism’s motility [56].

##### Nucleic Acid Detection Methods

Nucleic acid amplification methods allow for the detection of *T. vaginalis* DNA or RNA, even in samples with low pathogen concentrations. These tests do not require viable organisms and can be used with a variety of clinical specimens [55].

Sensitivity typically ranges from 85% to 100%, depending on the primers and target sequences employed. These values refer to clinical sensitivity, representing the test’s ability to correctly identify infected individuals based on patient-derived samples. Analytical sensitivity, referring to the minimal detectable DNA quantity, is also high in these methods but varies with the specific platform and protocol used. To enhance the sensitivity of this technique, the development of new primers is necessary. In 1992, Riley et al. identified a set of primers, TVA5 and TVA6, which target conserved regions of the *Trichomonas vaginalis* 18S ribosomal RNA (rRNA) gene. This region is frequently used due to its species-specific sequence and high copy number, which improve both the sensitivity and specificity of detection [57].

NAATs are highly sensitive and specific methods for detecting *T. vaginalis*, but their complexity can vary significantly depending on the platform used. While some assays, such as real-time PCR systems, require specialized equipment and trained personnel, others—like cartridge-based or automated NAATs—are designed for ease of use and are suitable for point-of-care settings. In 2011, the FDA (Food and Drug Administration) approved the Aptima *T. vaginalis* test (Hologic Gen-Probe, San Diego, CA, USA), which detects RNA using transcription-mediated amplification (TMA) and demonstrates approximately 95% sensitivity and 100% specificity. This assay is compatible with automated systems such as Gen-Probe’s TIGRIS platform (Hologic Inc., San Diego, CA, USA), facilitating rapid and high-throughput testing [51,59].

It represents an adaptation of the Aptima Combo 2 Chlamydia/Gonorrhoea method (Hologic, San Diego, CA, USA), allowing *T. vaginalis* diagnosis using samples collected for Chlamydia/Gonorrhoea detection [54].

Other FDA-approved NAATs include the BD ProbeTec TV Qx (BD Diagnostics, Sparks, MD, USA), which relies on strand displacement amplification (SDA) and offers a sensitivity of 98.3% and specificity of 99.6% [61]. This test is performed using the BD Viper system (Becton, Dickinson and Company, BD Diagnostics, Franklin Lakes, NJ, USA), an automated platform capable of processing sample preparation through nucleic acid extraction followed by amplification [62]. The BD Max CTGCTV2 (Becton, Dickinson and Company, Franklin Lakes, NJ, USA) is a multiplex real-time PCR assay that allows simultaneous detection of *Chlamydia trachomatis*, *Neisseria gonorrhoeae*, and *Trichomonas vaginalis*. It operates on the fully automated BD Max System (Becton, Dickinson and Company, Franklin Lakes, NJ, USA), integrating DNA extraction, amplification, and detection in a single workflow. This method is validated for both vaginal and urine samples and shows sensitivity and specificity values above 90% depending on the matrix. The Roche Cobas TV/MG test (Roche Molecular Systems, Inc., Pleasanton, CA, USA) also demonstrates robust performance across various sample types [61].

Although these methods are highly accurate, their implementation is constrained by high costs, the need for specialized laboratory infrastructure, and trained personnel. As such, they are not universally available, especially in low-resource settings. Nevertheless, their role in forensic diagnostics is increasingly relevant, particularly in cases where rapid and reliable confirmation of infection is necessary [53].

##### Methods Based on Oligonucleotide Probes

Oligonucleotide probe-based methods are also commercially available and can be performed in clinical settings, with a sensitivity of 80–90% and a specificity of 95% [57].

One such detection method utilizes synthetic DNA oligonucleotide probes, such as the BD Affirm VPIII (Becton Dickinson, Sparks, MD, USA), which has been approved by the U.S. Food and Drug Administration (FDA) for detecting *T. vaginalis*, *Gardnerella vaginalis*, and *Candida* species in a single reaction. This method demonstrates a sensitivity of 91–100% and a specificity of 93–96%, with results available in approximately one hour [61].

*T. vaginalis* has approximately eight serotypes, and antibody-based techniques, such as immunoblotting, have been studied. Methods including complement fixation, hemagglutination, gel diffusion, fluorescent antibodies, and ELISA have been employed to detect trichomoniasis antibodies. However, these techniques are not specific for detecting recent infections [56].

## 5. Rapid Point-of-Care Testing

POC tests provide a solution for obtaining rapid diagnoses outside of laboratory settings. These tests can be conducted in hospitals, clinics, or near-patient settings such as emergency rooms or shelters. While theoretically adaptable for use at crime scenes, their application in such contexts is highly limited due to strict legal and procedural requirements, including the need to preserve the chain of custody, ensure sample integrity, and comply with jurisdictional forensic protocols. Therefore, in practice, the use of diagnostic testing for STI detection typically occurs in clinical or forensic laboratory environments following proper evidence handling procedures. They must meet the WHO criteria, including sensitivity, accessibility, specificity, ease of use, speed, robustness, and the ability to function without specialized equipment [14].

For gonorrhoea, POC tests based on molecular polymerase chain reaction (PCR), such as GeneXpert NG/CT (Cepheid, Sunnyvale, CA, USA), have demonstrated superior diagnostic performance compared to laboratory-based NAATs. Other NAAT techniques utilizing isothermal amplification, instead of the thermal cycling required for PCR, help reduce costs and shorten the time needed to obtain results [15]. These DNA/RNA detection-based tests provide faster and more sensitive results than traditional microscopy and culture methods [14].

For the rapid detection of chlamydia, available tests include Clearview (Unipath Ltd., Bedford, UK), TestPack (Abbott, North Chicago, IL, USA), and SureCell (Johnson & Johnson, Rochester, NY, USA) [28]. More recently, the molecular test Xpert (Cepheid, Sunnyvale, CA, USA) has been developed, utilizing nucleic acid amplification techniques with high accuracy. Another advancement is the Io POC Chlamydia test (Atlas Genetics, Bath, UK), which employs PCR in microfluidic systems with electrochemical detection of results. Isothermal amplification tests, such as LAMP and RPA, generate faster results than PCR and enhance the use of NAATs in point-of-care settings [27].

For syphilis detection, the most commonly used rapid tests are immunochromatographic strips coated with recombinant *T. pallidum* antigens, SD Bioline Syphilis 3.0, designed to detect IgM, IgG, or IgA antibodies. These tests exhibit sensitivity and specificity similar to conventional treponemal tests but are unable to distinguish between past and current infections [34]. Their advantages include low cost, rapid turnaround time, ease of use without specialized personnel, and the potential for early syphilis detection [41].

Studies on available rapid treponemal tests have shown a sensitivity of 76–86% and a specificity of 96–99% compared to laboratory assays such as the TPPA [47].

Finally, for the rapid POC detection of trichomoniasis approved by the U.S. Food and Drug Administration (FDA), in addition to wet mount microscopy, lateral flow methods are also available. These include the OSOM^®^ Trichomonas Rapid Test (Sekisui Diagnostics, Bedford, MA, USA) and the AmpliVue helicase-dependent isothermal amplification test (Quidel, San Diego, CA, USA), which provides results within approximately 45–50 min [58], as well as the Solana TV test (Quidel, San Diego, CA, USA) [61].

The OSOM immunochromatographic enzyme assay (Sekisui, Framingham, MA, USA) utilizes a latex-labeled antibody that specifically binds to Trichomonas proteins. It also incorporates a secondary antibody to capture the formed complex within the lateral flow device, enabling detection. Results are obtained within approximately 15 min [54].

The Cepheid GeneXpert TV test (Cepheid, Sunnyvale, CA, USA) is an FDA-approved rapid test that provides results within one hour. Its sensitivity ranges from 99.5% to 100%, and its specificity is between 99.4% and 99.9% when compared to wet mount and culture methods [61]. This method is based on real-time PCR, with fully automated sample preparation [62].

Additionally, an antigen detection test has been developed for point-of-care use, facilitating trichomoniasis detection (Genzyme Corp., Cambridge, MA, USA). Compared to culture, this immunoenzymatic test demonstrates a sensitivity of 78.5% and a specificity of 98.6% [57].

Another detection method is the Kalon TV latex agglutination test (Kalon Biological, Surrey, UK), which has not yet been FDA-approved. This test can be conducted at the POC using latex spheres coated with antibodies specific to *T. vaginalis* proteins. Its advantages include not requiring specialized instruments and delivering results within approximately 10 min. Compared to wet mount and culture methods, this test has a sensitivity of 98.8% and a specificity of 92.1% [63].

To facilitate comparison between the different rapid point-of-care tests discussed in this section, the following Table 2 summarizes their main characteristics, including the type of diagnostic method, time to result, diagnostic performance, and regulatory approval status. This overview highlights the practicality and diagnostic value of these tools in clinical and forensic contexts.

### 5.1. Multiplex Detection of STIs

In addition to single-target diagnostic tests, several multiplex molecular platforms have been developed to enable the simultaneous detection of multiple STIs within a single sample. These include assays such as the Cepheid Xpert CT/NG, BD MAX CT/GC/TV, and Aptima Combo 2, which utilize NAATs to concurrently detect *Chlamydia trachomatis*, *Neisseria gonorrhoeae*, and *Trichomonas vaginalis*, among others. These systems are particularly beneficial in syndromic management, where overlapping symptoms do not allow for reliable pathogen-specific diagnosis.

Multiplex NAATs exhibit high clinical sensitivity and specificity and are especially valuable in forensic and clinical settings requiring broad STI screening. However, their application is limited in low-resource environments due to their cost, infrastructure demands, and requirement for trained laboratory personnel. Despite these constraints, multiplex NAATs currently represent the most efficient option for multi-pathogen STI detection, and continued innovation in portable, microfluidic-based platforms may further expand their accessibility and utility in field settings.

### 5.2. Comparative Reflection on Diagnostic Strategies

Building upon the diagnostic methods detailed in the previous sections, a comparative reflection is warranted to contextualize their applicability, strengths, and limitations. This mid-article synthesis aims to critically assess the suitability of each method across different infections and scenarios.

When comparing diagnostic techniques across the four major STIs reviewed—gonorrhoea, chlamydia, syphilis, and trichomoniasis—it becomes evident that no single method offers universal applicability. Instead, diagnostic suitability is determined by factors such as the pathogen’s biology, infection stage, specimen type, and testing context (clinical vs. forensic).

For example, NAATs (including PCR and TMA) show consistently high clinical and analytical sensitivity across all four STIs, making them the most robust option in terms of diagnostic accuracy. However, their reliance on specialized equipment and trained personnel limits their implementation in low-resource or point-of-care settings.

Culture methods, though specific and useful for antimicrobial susceptibility testing (particularly in *Neisseria gonorrhoeae*), are limited by the need for viable organisms, stringent transport conditions, and longer turnaround times. Similarly, microscopy, while rapid and cost-effective, suffers from low sensitivity, especially in asymptomatic cases or for pathogens with low organism load such as *Chlamydia trachomatis*.

For syphilis, the combination of treponemal and non-treponemal tests remains the gold standard, yet interpretation can be complicated due to lifelong seropositivity in treponemal assays and false-positives in non-treponemal methods.

Point-of-care tests, particularly those based on immunochromatography or isothermal amplification, represent a promising compromise between speed and practicality, although variability in sensitivity—especially for antibody-based tests—remains a concern.

In summary, NAATs emerge as the most reliable diagnostic method overall due to their versatility, high sensitivity/specificity, and ability to detect co-infections. Nonetheless, POC methods are essential for rapid, initial screening—especially in forensic and emergency contexts—where laboratory access is limited. Tailoring the diagnostic strategy to the specific clinical and forensic context, while considering the pathogen’s characteristics, remains essential for optimizing STI detection in victims of sexual violence.

This comparative overview underscores the need for context-aware diagnostic choices and provides a framework for the concluding discussion that follows.

## 6. Discussion

Asymptomatic infections and the limitations of symptom-based diagnosis are highly prevalent in sexually transmitted infections (STIs), highlighting the necessity of laboratory confirmation for accurate detection [14]. This step is essential to ensure appropriate treatment for affected individuals and to prevent the further spread of these diseases. To address certain challenges, nucleic acid tests and non-invasive, close-to-patient diagnostic tools have been developed in recent years. These tests can be self-administered without the need for an STI specialist, performed in various settings, and processed at the point of care, delivering results within minutes or hours [10].

Clinical symptoms alone are rarely sufficient to accurately identify a specific STI, making laboratory diagnosis indispensable for confirming the type of infection present [14]. However, careful consideration must be given to the limitations of each test, as supplementary diagnostic methods may sometimes be required to verify results. When a test is conducted for medico-legal purposes, such as in cases of sexual assault or to establish a link between a victim and a perpetrator, maintaining a proper chain of custody is crucial to ensure the integrity of the evidence [10].

In addition to microbiological testing, DNA profiling plays a central role in the forensic investigation of sexual assault cases. It provides a highly specific form of evidence capable of directly linking a suspect to a victim or crime scene, particularly when biological traces such as semen, blood, or epithelial cells are recovered. While the detection of STIs may support the occurrence of sexual contact—especially in cases where the same pathogen is identified in both victim and suspect—STI testing alone cannot establish the source or timing of transmission. Therefore, DNA analysis offers stronger probative value and is considered the gold standard in forensic identification. Nonetheless, STI detection can serve as a complementary tool, especially in cases where DNA evidence is unavailable, degraded, or insufficient due to delayed reporting. The integration of both approaches enhances the robustness of forensic interpretation, particularly in complex scenarios involving asymptomatic infections or vulnerable populations such as children.

The selection of recommended diagnostic tests depends on various factors, including the victim’s circumstances, availability of resources, individual capability, age, type of assault, and the time elapsed since the incident [10].

Choosing the most appropriate diagnostic method is not solely determined by its sensitivity, specificity, and predictive values but also by logistical considerations such as equipment availability, cost-effectiveness, and performance, as well as the intended purpose of the test. In resource-limited settings, the use of highly sensitive and specific tests is often impractical due to their high costs and equipment requirements. Therefore, rapid diagnostic methods are essential to enable immediate treatment for patients [14].

This article has outlined various detection methods developed to diagnose the most prevalent sexually transmitted infections (STIs) worldwide. Traditional microscopy and culture techniques present several disadvantages, including the necessity of examining the sample within 10 min of collection [14], the requirement for viable bacteria [29], reliance on specialized equipment such as microscopes, and the need for trained personnel. These factors make the implementation of such methods challenging in resource-limited areas [58].

To overcome these limitations, alternative diagnostic techniques have been developed, including nucleic acid detection methods, polymerase chain reaction (PCR)-based techniques, serological tests, and rapid POC tests. These POC methods offer a viable solution for the rapid diagnosis of STIs outside of laboratory settings [14]. Additionally, they provide advantages in sample collection, as traditional laboratory-based methods can be invasive. Lower equipment costs also facilitate the establishment of diagnostic facilities closer to patients, increasing accessibility to testing and treatment [29].

Looking ahead, the development of advanced diagnostic technologies remains a priority. Faster diagnostic tools with forensic capabilities are crucial in cases of sexual violence. Innovations in these methods could enhance accuracy and reduce the likelihood of false-negatives. Moreover, the development of non-invasive sample collection techniques, such as saliva or capillary blood sampling, could serve as alternatives or complements to current genital sample collection methods. Such advancements would help minimize the trauma experienced by victims while improving compliance in forensic investigations and medical diagnoses.

These future perspectives emphasize the importance of integrating technological advancements in diagnostics with enhanced collaboration between healthcare services and judicial authorities. By fostering progress in STI detection, healthcare improvements, and victim support systems, it is possible to envision a future where both prevention and care for STI victims are strengthened, contributing to a safer and more just society.

From the authors’ perspective, the integration of rapid, reliable, and minimally invasive diagnostic tools into forensic protocols is not merely a technical advancement, but a fundamental step toward more victim-centered forensic practice. In the context of sexual violence, where timely intervention and trauma-sensitive approaches are essential, point-of-care methods offer an opportunity to improve both clinical care and the quality of forensic evidence. However, the implementation of such tools must be accompanied by standardized guidelines, professional training, and strengthened cooperation between healthcare providers and judicial authorities. We believe that diagnostic innovation, when aligned with ethical and legal frameworks, has the potential to significantly enhance the support system for victims of sexual crimes and contribute to a more effective and humane justice process.

### 6.1. Forensic Relevance and Multiplexing Considerations

Although many of the diagnostic techniques described in this review were developed and validated for clinical settings, their application in forensic contexts presents distinct challenges. In cases of sexual violence, the primary objective of STI detection extends beyond medical treatment; it also includes the potential to support legal proceedings by establishing biological links between the victim and the alleged perpetrator. Therefore, forensic diagnostic tools must adhere to rigorous standards of sample integrity, chain of custody, reproducibility, and evidentiary admissibility. Methods that require complex equipment or involve highly sensitive molecular reactions, such as multiplex NAATs, may be difficult to implement in forensic laboratories with limited infrastructure or where test results must be easily interpretable in court.

While multiplex platforms—capable of detecting multiple pathogens in a single sample—offer practical and diagnostic advantages (particularly for asymptomatic victims or co-infections), they also raise concerns regarding data complexity, legal interpretation, and validation for forensic use. The lack of standardization and variation in performance across test brands may further limit their forensic applicability. Moreover, these tests rarely provide information on time of infection, which is often a critical element in forensic analysis.

Therefore, despite their diagnostic potential, multiplex methods must be carefully evaluated for forensic use. Future research should focus on the development of multiplex-compatible forensic protocols, including robust chain-of-custody documentation, validation studies in forensic scenarios, and interdisciplinary guidelines that integrate medical diagnostics into judicial processes with reliability and ethical rigor.

### 6.2. Study Limitations

This narrative review provides an overview of current diagnostic approaches for sexually transmitted infections in the context of sexual violence. However, several limitations should be acknowledged. First, the non-systematic nature of the literature review may introduce selection bias, as formal inclusion and exclusion criteria were not applied. Second, the scarcity of forensic-oriented studies focusing on the detection of STIs specifically within the context of sexual crimes limits the depth of analysis concerning medico-legal applicability. Third, the absence of original data collection constrains the ability to compare diagnostic accuracy across different methods in real-world settings. Moreover, variations in healthcare infrastructure, access to diagnostic tools, and laboratory capabilities across different countries may reduce the generalizability of the findings. These limitations highlight the need for further empirical research and standardization of STIs diagnostic protocols, particularly in forensic and resource-limited environments.

## 7. Conclusions

The literature review revealed that sexual crimes can be associated with the transmission of sexually transmitted infections (STIs), including gonorrhoea, chlamydia, syphilis, and trichomoniasis. Early detection of these STIs plays a crucial role in ensuring not only appropriate medical treatment for victims but also in preventing long-term health consequences and the further spread of these infections.

STIs in victims of sexual crimes represent a complex issue, requiring responses that address both clinical and forensic needs. This review explored the primary STI detection methods within the context of sexual violence, analyzing the challenges, advancements, and limitations of current diagnostic methodologies. New approaches, such as POC methods, have emerged as promising solutions to enhance the speed and accuracy of diagnosis.

Despite significant progress in STI detection techniques, there remain gaps in the effectiveness and accessibility of newer diagnostic methods. Therefore, further research is essential to explore and develop innovative technologies and strategies that optimize the detection and treatment of STIs in cases of sexual violence.

Continued research efforts should focus on improving diagnostic methods to make them less invasive, faster, and more applicable in forensic contexts. Additionally, the implementation of public policies and educational campaigns is vital to raise awareness of STIs, promote timely medical care, and reduce stigma. Enhancing the training of healthcare and forensic professionals, alongside the development of ethical and practical guidelines, will ensure that forensic and clinical care for victims of sexual crimes is conducted with both responsibility and effectiveness.

Ultimately, future studies will contribute to the advancement of early detection methods and the development of new victim care protocols. Strengthening the integration between healthcare and the justice system is fundamental to improving the protection and support available to victims, ensuring a more comprehensive and effective response to sexual violence.

## Figures and Tables

**Figure 1 diagnostics-15-01294-f001:**
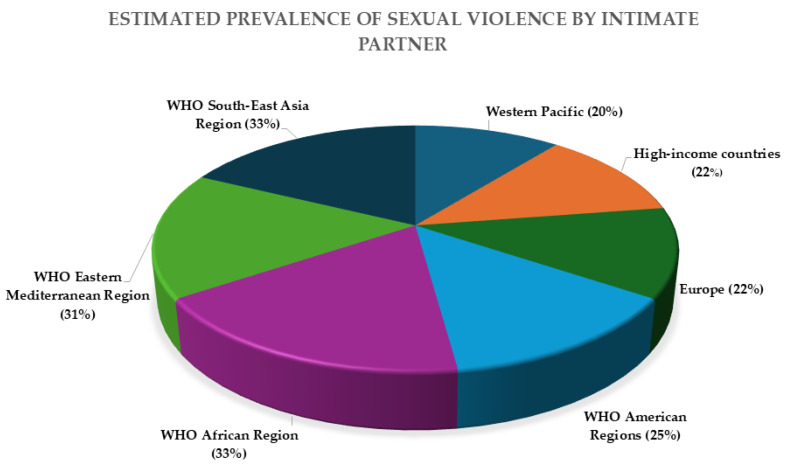
Estimated lifetime prevalence of intimate partner violence in different regions of the world (adapted from the [3]).

**Figure 2 diagnostics-15-01294-f002:**
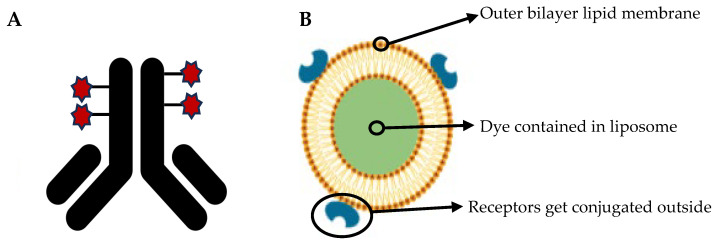
The structure of the liposomal nanovesicles and detection antibody. (**A**) Detection antibody conjugated with biotin. (**B**) Liposomal nanovesicle (adapted from [31]).

**Figure 3 diagnostics-15-01294-f003:**
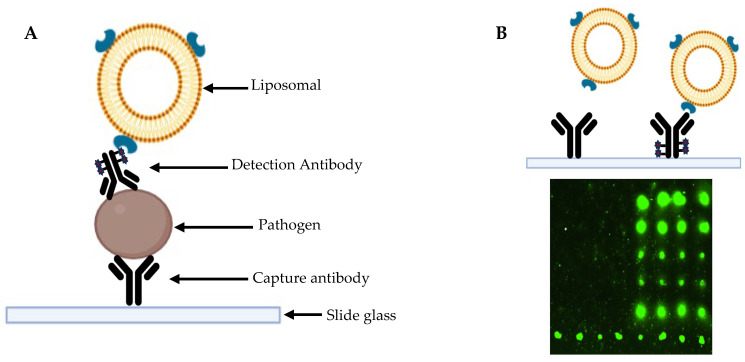
Microorganism detection using a microarray chip and representative examples: (**A**) Overview of the detection process; (**B**) illustrative systems utilizing either native or antibody-conjugated glass surfaces, with positive detection demonstrated in the latter (adapted from [31]).

**Table 1 diagnostics-15-01294-t001:** Estimated prevalence of major STIs by WHO region. This table presents the estimated prevalence of the four most common sexually transmitted infections—syphilis, chlamydia, gonorrhoea, and trichomoniasis—across different WHO regions, adapted from WHO data (adapted from [3]).

WHO Region	Syphilis (%)	Chlamydia (%)	Gonorrhoea (%)	Trichomoniasis (%)
Africa	3.5	6.0	2.9	11.0
Americas	0.5	4.2	0.7	3.0
Eastern Mediterranean	1.2	3.8	0.9	4.1
Europe	0.3	3.5	0.4	1.2
South-East Asia	0.7	5.0	1.0	5.3
Western Pacific	0.4	4.0	0.6	2.8

**Table 2 diagnostics-15-01294-t002:** Table summarizes their main characteristics, including the type of diagnostic method, time to result, diagnostic performance, and regulatory approval status.

Infection	Test Name	Method	Time to Result	Sensitivity (%)	Specificity (%)	FDA Approved
Gonorrhoea	GeneXpert NG/CT	Real-time PCR	~90 min	95–100	98–100	Yes
Chlamydia	Io POC Chlamydia	PCR (microfluidic)	~30 min	~96	~99	Yes
Syphilis	SD Bioline Syphilis 3.0	Immunochromatographic strip	~20 min	76–86	96–99	Yes
Trichomoniasis	OSOM^®^ Trichomonas Rapid Test	Immunochromatography	~15 min	~95	~97	Yes
	AmpliVue TV	Isothermal amplification	~50 min	~98	~99	Yes
	Cepheid GeneXpert TV test	Real-time PCR	~60 min	99.5–100	99.4–99.9	Yes

## Data Availability

No new data were created or analyzed in this study.

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
