# Peer review of "Forensic Microbiology: Challenges in Detecting Sexually Transmitted Infections"

_diagnostics, 2025, doi:10.3390/diagnostics15101294_

Round 1

Reviewer 1 Report

Comments and Suggestions for Authors

I congratulate the Authors on their work; I found it interesting and well-designed.

Here are some points that could be improved from my perspective:

Abstract: Consider moving lines 26–29 to follow lines 30–34, as they refer to the conclusion and would logically come after the study’s aim.

Methods: Move lines 87–88 to the beginning of the Methods section for better flow.

Figures 1 and 2: These appear to be inverted. Figure 1 corresponds to section 4, while Figure 2 relates to section 3.

Section numbering: Points 5, 6, 7, and 8 address infections and currently appear as subsections under point 4, "Sexually Transmitted Infections (STIs)." Please consider renumbering them as 4.1, 4.2, 4.3, and 4.4, or choose another structural design for clarity.

Section 9: Since the study aims to analyze "rapid point-of-care methods," I suggest giving section 9 greater prominence. You might add figures or tables illustrating the different tests and their characteristics. Additionally, consider dividing this section into subparagraphs for each type of infection.

Limitations: Please add a paragraph discussing the limitations of the study.

Reviewer 2 Report

Comments and Suggestions for Authors

The article well reviewed present detection methods of four typical sexually transmitted infections. The reviewer has no special opinion on this article except for one minor point. After correction, the article should be accepted.

Comment

To evaluate the performance of each test, the authors used sensitivity and specificity. The reviewer thinks that these indices cannot be used without presence of gold standard. Please clarify what test is gold standard.

Reviewer 3 Report

Comments and Suggestions for Authors
  • The topic of detecting sexually transmitted infections (STIs) is highly engaging and relevant.
  • The English used in the article is good and makes for pleasant reading.
  • The review remains quite superficial and lacks depth.
  • The description of symptoms for various STIs is disproportionately extensive, especially in Chapter 7, where all forms of syphilis are discussed. This could be condensed into a single section for better readability.
  • The short paragraphs and additional white spaces from Section 4 onwards make the text appear longer.
  • Essential details about the methods used are missing. Questions like sensitivity, specificity, causes of false positives, duration, and sample types (invasive or non-invasive) are not addressed.
  • Surprisingly, for an article with 'forensic' in the title, the introduction barely touches on forensic aspects. Sections 5 to 8 do not mention forensics at all.
  • Be consistent in terminology. For example, PCR-based assays, molecular techniques, NAAT. In R334, the abbreviation NAA is used for nucleic acid amplification (NAA) test, and in R336, NAAT is used for nucleic acid amplification testing. Use microscopy as a general term instead of dark field microscopy (7.7.3), microscopy (5.1.1), and wet mount microscopy (8.1.1). The distinction between different techniques follows from the text under the heading.
  • Some terms may need a brief explanation, such as 'vertically transmitted', Vitek2 (what type of method is this?), what is included under NAAT/what does it mean?, (non-)treponemal.
  • Including explanatory figures of what such a test looks like would make the article more readable. For example, in Section 6.1.4, curiosity is piqued about what such a test looks like.
  • Specify whether the mentioned values are clinical sensitivity (what it is) or analytical sensitivity.
  • A mid-article conclusion/summary is missing, which should include a critical reflection on existing methods. Why are some methods suitable or not for certain STIs? Which method is the best and why? Chapter 9 touches on this but could be more critical and reflective. This would add value to the review rather than just listing existing literature.
  • Is there a test that works for all/multiple STIs?
  • The discussion focuses on medical aspects rather than forensic ones. Why then is 'forensic' in the title? The discussion is too generic; it lacks depth and critical reflection. Also, delve more into the (im)possibilities of multiplexing.

Specific Points:

  • R100-102: The text describing the figure does not match the figure itself.
  • Section 3: Contains a lot of repetition and could be more concise.
  • Table 1: Could be removed or moved to the appendix. It could also be more clustered (e.g., unwanted pregnancy with abortion).
  • R118: In sexual assault cases, DNA profiling is also widely used. More information on this and its relation to STI detection would be beneficial. DNA profiling, if possible, is a much stronger evidence.
  • There is a lot of discussion about the prevalence of STIs, but how often is this related to sexual assault?
  • Figure 2: Only addresses cases of sexual violence involving intimate partners. How does this figure look for non-intimate partners?
  • Sections 5-8: These sections could be much shorter. An introduction stating that symptoms (please briefly) for each STI will be described first, followed by detection methods, is missing.
  • R225: Detection without amplification, is it then NAAT?
  • R283: Which sample?
  • R311: Multiplex for what?
  • Table 2: Contains too much detail and could be removed (as this level of detail is not provided for other STIs) or moved to an appendix.
  • R593: Why are there two chambers?
  • R624: Are NAATs generally complex or just this specific one?
  • R667: Are these tests actually used at crime scenes? What about legal regulations?
  • For readability, avoid paragraphs of 1-3 lines, as frequently seen in Section 8.1.4.

Reviewer 4 Report

Comments and Suggestions for Authors

Dear Authors,
The article is interesting and provides a wealth of valuable information. However, several techniques are mentioned without being adequately described, and some abbreviations appear in the text without explanation, even though they are clarified in a separate section. Furthermore, in some paragraphs the abbreviations are explained, while in others they are not. This can make it challenging for readers to fully understand the content.  In my opinion it would benefit from revision according to the following suggestions:

-2. Methods

Indicate the period the articles refer to. Clarify the exclusion criteria.

-5.1.1. Microscopy

Line 158: Describe the principle and procedure of Gram staining to enhance the reader's understanding.

Line 185: Clarify what the Vitek 2 method consists of

- 5.1.3. Nucleic acid amplification tests

Line 194: Add (NAAT)  beside the paragraph heading

-5.1.4. Immunological tests

Line 244:  Include sexually transmitted infections instead of STIs and put STIs in parentheses.

- 6.1.1. Culture

Line 272: clarify whta  HeLa and BGMK mean

-6.1.3. Serological test

Line 307 include lipopolysaccharide and put LPS in parentheses.

-6.1.5. Molecular techniques

Line 326: explicate in what PACE 2 test consists.

-7.7.5. PCR

Title: Replace PCR with Polymerase Chain Reaction and put PCR in parentheses.

Line 469: clarify the concept: Real-time PCR allows for the quantification of pathogens based on a standard curve, enabling an estimation of infection severity.

-8.1.4. Nucleic Acid Detection Methods

Line 623: Clarify which regions the primers target

Line 629: include Manufacturer for Gen-Probe and TIGRIS

Line 639: include Manufacturer for BD Viper system

Line 644: explicate in what BD Max CTGCTV2 consists

-10. Discussion

The authors' perspective on the discussed topic should be more clearly highlighted.

Finally, Fig. 1 may be transformed in a table, otherwise the figure may be included as supplementary data.

Table 1 and 2 should be resized.

Round 2

Reviewer 3 Report

Comments and Suggestions for Authors

I would like to thank the authors for thoroughly addressing the feedback provided. The manuscript has become significantly more readable and its academic value has clearly improved. It is good to see that the first part has been made more concise, while the second part now offers greater depth. I appreciate the suggestion to group all STIs under one chapter; however, I find that the introduction of five levels of subheadings does not necessarily improve readability. While the paper has been considerably strengthened, the forensic context could still be sharpened further, particularly in the introduction and early sections.

Author Response

Comments 1: I would like to thank the authors for thoroughly addressing the feedback provided. The manuscript has become significantly more readable and its academic value has clearly improved. It is good to see that the first part has been made more concise, while the second part now offers greater depth. I appreciate the suggestion to group all STIs under one chapter; however, I find that the introduction of five levels of subheadings does not necessarily improve readability. While the paper has been considerably strengthened, the forensic context could still be sharpened further, particularly in the introduction and early sections.

Response 1: 

Dear Reviewer,

We would like to sincerely thank you for your thoughtful and constructive comments, which have contributed significantly to improving the clarity and scientific value of our manuscript.

We fully understand your observation regarding the use of five levels of subheadings and agree that this may impact the overall readability. However, at this stage of the revision process, restructuring the manuscript to reduce the subheading levels would require substantial modifications that could compromise the coherence and intended flow of the text. We truly appreciate your feedback and will carefully consider this important aspect in future manuscripts.

Regarding your suggestion that the forensic context could be further strengthened, particularly in the introduction and early sections, we have revised the text accordingly to provide a clearer and more explicit forensic framework. 

The suggested improvements regarding the forensic context have been addressed and incorporated into the revised introduction. Specifically, these changes can be found in lines 44 to 50 and 54 to 55, where we have emphasized the forensic relevance of sexually transmitted infections in the context of sexual crimes.

The modifications have been highlighted in blue for easier identification.

We hope these changes address your concern and meet your expectations.

Once again, thank you for your valuable contribution to the refinement of our work.

Kind regards,